# Information Design for Multiple Independent and Self-Interested Defenders: Work Less, Pay Off More

**Chenghan Zhou**[1]       **Andrew Spivey**[2]       **Haifeng Xu**[3]       **Thanh Hong Nguyen**[2]

[1]Computer Science Department, University of Virginia, Charlottesville, Virginia, USA
[2]Computer Science Department, University of Oregon, Eugene, Oregon, USA
[3]Department of Computer Science, University of Chicago, Chicago, Illinois, USA

## Abstract

This paper studies the problem of information design in a general security game setting in which multiple independent self-interested defenders attempt to provide protection simultaneously on the same set of important targets against an unknown attacker. A principal, who can be one of the defenders, has access to certain private information (i.e., attacker type) whereas other defenders do not. We investigate the question of how that principal, with additional private information, can influence the decisions of the defenders by partially and strategically revealing her information. In particular, we develop a polynomial-time ellipsoid algorithm to compute an optimal private signaling scheme. Our key finding is that the separation oracle in the ellipsoid approach can be carefully reduced to bipartite matching. Furthermore, we introduce a compact representation of any ex-ante persuasive signaling schemes by exploiting intrinsic security resource allocation structures, enabling us to compute an optimal scheme significantly faster. Our experiment results show that by strategically revealing private information, the principal can significantly enhance the protection effectiveness on the targets.

## 1 INTRODUCTION

In many real-world security domains, there are often multiple self-interested security teams who conduct patrols over the same set of important targets without coordinating with each other [Jiang et al., 2013]. Among others, an important motivating domain of this paper is wildlife conservation — while patrol teams from various NGOs or provinces patrol within the same conservation area to protect wildlife from poaching. Different NGOs or provinces typically have different types of targeted species (e.g., the situation in Pakistan [ministry of WWF-Pakistan, 2015]) and tend to operate separately. Similarly, there are multiple different countries which simultaneously plan their own anti-crime actions in international waters against illegal fishing [Klein, 2017].

The study of multi-defender security games has attracted much recent attention. Unfortunately, most findings so far are relatively *negative*. Specifically, [Lou et al., 2017] show that the lack of coordination among defenders may significantly lessen the overall protection effectiveness, leading to *unbounded* price of anarchy. In addition, [Gan et al., 2018] recently show that finding a Nash-Stackelberg equilibrium among the defenders, taking into account strategic response of the attacker, is computationally *NP-hard*. Given these negative results, this paper asks the following question:

> *How to obtain defense effectiveness and computation efficiency in multi-defender security games?*

To answer the above question, we exploit the use of *information* as a "knob" to coordinate strategic agents' decisions. Specifically, we study how a principal with privileged private information can influence the decisions of all defenders by strategically reveal her information, a task also known as *information design* or *persuasion* [Dughmi, 2017]. Concretely, we study information design in a Bayesian security game setting with multiple independent and self-interested defenders. These defenders attempt to protect important targets against an unknown attacker. The attacker *type* is unknown to the defenders. Nevertheless, all defenders share a common knowledge of a prior distribution over the attacker types. In this setting, there is a principal who has additional information about the attacker type and wants to communicate with both the defenders and the attacker through a persuasion signaling mechanism in order to influence all of their decisions towards the principal's goal. In wildlife protection, for example, the principal may be the national park office. Since many poachers (or the attacker) are local villagers, park rangers can have access to private information through local informants about whom (i.e., which attacker type) is conducting poaching [Viollaz and Gore, 2019].

*Accepted for the 38th Conference on Uncertainty in Artificial Intelligence* (UAI 2022).

In summary, our results show that information design not only significantly improves protection effectiveness but also leads to efficient computation. Concretely, assuming the principal can communicate with defenders privately (a.k.a., private signaling [Dughmi and Xu, 2017]), we develop an ellipsoid-based algorithm in which the separation oracle component can be decomposed into a polynomial number of sub-problems, and each sub-problem reduces to a bipartite matching problem. We remark that this by no means is an easy task, neither conceptually nor technically, since the outcomes of private signaling form the set of Bayes correlated equilibria [Bergemann and Morris, 2016] and computing an optimal correlated equilibrium is a fundamental and well-known intractable problem [Papadimitriou and Roughgarden, 2008]. Our proof is technical and crucially explores the special structure of security games. In addition, we also investigate the ex-ante private signaling scheme (a relaxation of private signaling in which the defenders and attacker decides whether to follow the principal's signals or not before any signal is realized [Castiglioni et al., 2021]). In this scenario, we develop a novel compact representation for the principal's signaling schemes by compactly characterizing jointly feasible marginals. This finding enables us to significantly reduce the signaling scheme computation.

Finally, we present extensive experiment results evaluating our proposed algorithms in various game settings. We evaluate two different principal objectives: (i) maximizing the defenders' social welfare; and (ii) maximizing her own utility. Our results show that through signaling schemes, the principal can significantly increase the social welfare of the defenders while substantially reducing the attacker's utility.

## 1.1 COMPARISON WITH PREVIOUS WORKS

**Security Games.** Security games refer to a well-studied class of games which capture strategic interactions between defenders and attackers in security domains [Tambe, 2011], with important real-world applications in, e.g., airport security [Pita et al., 2008], ferry protection [Shieh et al., 2012], and wildlife conservation [Fang et al., 2016]. Most relevant to our work is the recent study of multiple-defender security games. Several of them consider defenders to have identical interests [Jiang et al., 2013, Basilico et al., 2017] or to have their own disjointed set of targets [Laszka et al., 2016, Lou et al., 2017, Lou and Vorobeychik, 2016, 2015, Smith et al., 2014]. The game model in [Gan et al., 2018] is the most related to ours. This previous work investigates the existence and computation of a Nash-Stackelberg equilibrium among the defenders. To our knowledge, our work is the first to study information design in multi-defender security games. In contrast to previous negative results, our findings are much more encouraging. Our positive results even extend to more realistic game models with defender patrolling costs, which cannot be handled by the existing work.

**Information Design.** Information design, a.k.a. signaling, has attracted much interest in various domains such as public safety [Xu et al., 2015, Rabinovich et al., 2015], wildlife conservation [Bondi et al., 2019], traffic routing [Vasserman et al., 2015, Castiglioni et al., 2021] and auctions [Li and Das, 2019, Emek et al., 2012]. Most related to us is [Xu et al., 2016] which study signaling in Bayesian Stackelberg games. All previous work assumes a single defender whereas our paper tackles the complex *multiple-defender* setup. This requires us to work with exponentially large representation of signaling schemes and necessitates novel algorithmic techniques with compact representations.

**Other Learning-based Solutions.** Recent research in multi-agent reinforcement learning studies factors that influence agents' behavior in a shared environment. For example, [Tian et al., 2020] studies how to convey private information through actions in cooperative environment. [Jaques et al., 2019] uses monetary reward (which they call causal inference reward) to influence opponents' actions. Unlike the tools studied in previous MARL literature, our model takes advantage of information asymmetry to influence attackers' actions in an adversarial environment. Therefore, both our setup and approach are different from these previous learning-based methods.

## 2 PRELIMINARY

We consider a general security game setting in which there are multiple self-interest defenders, $\mathbf{D} = \{1, \dots, |\mathbf{D}|\}$, who have to protect important targets $\mathbf{T} = \{1, \dots, |\mathbf{T}|\}$ from an attacker. Each defender can protect at most 1 target.[1] The defenders do not know the attacker's type, but share common prior knowledge about the distribution over possible attacker types, $\{q(\lambda)\}_{\lambda \in \Lambda}$ with $\Lambda = \{1, \dots, |\Lambda|\}$, where $q(\lambda)$ is the probability that the attacker has type $\lambda$. If a defender $d$ decides to go to a target $t$, he has a patrolling cost of $C^d(t) < 0$. If the attacker $\lambda$ successfully attacks a target $t$, it receives a reward $R^\lambda(t) \geq 0$ while each defender $d$ receives a penalty $P^d(t) \leq 0$. Conversely, if any of the defender catches the attacker $\lambda$ at $t$, the attacker receives a penalty $P^\lambda(t) < 0$ while each defender $d$ obtains a reward $R^d(t) > 0$. Notably, one defender suffices to fully protect a target whereas multiple defenders on the same target will *not* be any more effective. This is the major reason of inefficiency without coordination [Lou et al., 2017].

## 3 OPTIMAL PRIVATE SIGNALING

We first study the design of private signaling schemes which help the principal to coordinate the defenders. The principal leverages her private information about the attacker type to

---

[1] This is w.l.o.g since any defender who can cover multiple targets can be "split" into multiple defender with the same utilities.

influence the decisions of all players (including the attacker) by strategically revealing her information. We adopt the standard assumption of information design [Kamenica and Gentzkow, 2011], and assume that the principal commits to a signaling scheme $\omega$ and $\omega$ is publicly known to all players. At a high level, a private signaling scheme generates a random variable called *signal profile* $\mathbf{s}$, which is correlated with $\lambda$, where $s(d)$ is the private signal sent to the defender $d$ and $s(a)$ is the signal sent to the attacker. Each defender $d$, once receiving a certain private signal $s_0$, updates his belief on the attacker type, using Bayes rule as follows:

$$P(\lambda \mid s_0) = \frac{q(\lambda) \sum_{\mathbf{s}:s(d)=s_0} \omega(\mathbf{s} \mid \lambda)}{\sum_{\lambda'} q(\lambda') \sum_{\mathbf{s}:s(d)=s_0} \omega(\mathbf{s} \mid \lambda')}$$

where $\omega(\mathbf{s} \mid \lambda)$ is the probability the signal profile $\mathbf{s}$ is generated given the attacker type is $\lambda$.

Any private signaling scheme induces a Bayesian game among players. According to [Bergemann and Morris, 2016], all the Bayes Nash equilibria that can possibly arise at any private signaling scheme forms the set of *Bayes correlated equilibrium* (BCEs). Similar to the standard correlated equilibria, the signals of a private signaling scheme in a BCE can also be interpreted as *obedient* action recommendations. Therefore, a private signal profile can be represented as $\mathbf{s} = (\{s(d)\}, s(a))$ where $s(d) \in \mathbf{T}$ is the suggested protection target for defender $d$ and $s(a) \in \mathbf{T}$ is the suggestion of a target to attack for the attacker. With slight abuse of notations, we use $s(-a)$ to represent the set of signals sent to the defenders and $s(-a, -d)$ is the set of signals sent to other defenders except the defender $d$.

## 3.1  AN EXPONENTIAL-SIZE LP FORMULATION

Like typical formulation of optimal correlated equilibrium, optimal private signaling can also be formulated as an exponentially large linear program (LP). Specifically, the principal attempts to find an optimal signaling scheme $\Omega = \{\omega(\mathbf{s} \mid \lambda)\}$ to optimize her objective, which can be either her own utility (if she is a defender) or the social welfare of the defenders. We abstractly represent the principal's objective function w.r.t a signal $\mathbf{s}$ as $U(\mathbf{s})$. The optimal private signaling can be formulated as following LP:

$$\max \sum_{\lambda} q(\lambda) \sum_{\mathbf{s}} \omega(\mathbf{s} \mid \lambda) U(\mathbf{s}) \text{ s.t.} \tag{1}$$

(Attacker obedience) $\forall \lambda, t, t'$ :

$$\sum_{\mathbf{s}:t=s(a)} \omega(\mathbf{s} \mid \lambda) U^{\lambda}(\mathbf{s}) \geq \sum_{\mathbf{s}:t=s(a)} \omega(\mathbf{s} \mid \lambda) U^{\lambda}(s(-a), t') \tag{2}$$

(Defender obedience) $\forall d, t, t'$ :

$$\sum_{\lambda} q(\lambda) \sum_{\mathbf{s}:s(d)=t} \omega(\mathbf{s} \mid \lambda) U^d(\mathbf{s}) \tag{3}$$

$$\geq \sum_{\lambda} q(\lambda) \sum_{\mathbf{s}:s(d)=t} \omega(\mathbf{s} \mid \lambda) U^d(s(-a, -d), t', s(a))$$

$$\sum_{\mathbf{s}} \omega(\mathbf{s} \mid \lambda) = 1, \omega(\mathbf{s} \mid \lambda) \geq 0, \forall \mathbf{s}, \lambda \tag{4}$$

where (2–3) are obedience constraints which guarantee the attacker of any type and all defenders will follow the principal's recommendation. The utilities of each defender $d$ and each attacker type $\lambda$ are determined as follows:

$$U^d(\mathbf{s}) = C^d(s(d)) + P^d(s(a)), \text{ if } \forall d' : s(a) \neq s(d')$$
$$U^d(\mathbf{s}) = C^d(s(d)) + R^d(s(a)), \text{ if } \exists d' : s(a) = s(d')$$
$$U^{\lambda}(\mathbf{s}) = R^{\lambda}(s(a)), \text{ if } \forall d' : s(a) \neq s(d')$$
$$U^{\lambda}(\mathbf{s}) = P^{\lambda}(s(a)), \text{ if } \nexists d' : s(a) = s(d')$$

Problem $(1 - 4)$ has an exponential number of variables $\{\omega(\mathbf{s} \mid \lambda)\}$ due to exponentially many possible defender allocations. This is also the common challenge in computing optimal correlated equilibrium for succinctly represented games with many players (defenders in our case). Indeed, optimal correlated equilibrium is proved to be NP-hard in many succinct games [Papadimitriou and Roughgarden, 2008]. Perhaps surprisingly, next we show that LP $(1 - 4)$ can be solved in a polynomial time in our case.

## 3.2  A POLYNOMIAL-TIME ALGORITHM

We prove the following main positive result.

**Theorem 1.** *The optimal private signaling scheme can be computed in polynomial time.*

The rest of this section is devoted to the proof of Theorem 1. We elaborate the proof for the principal objective of maximizing the defender social welfare, i.e., $U(\mathbf{s}) = \sum_d U^d(\mathbf{s})$. The proof is similar when the principal is one of the defender. Our proof is divided into three major steps, and crucially exploits the structure of security games.

**Step 1:** Restricting to simplified pure strategy space. One challenge of designing the signaling scheme is when multiple defenders are recommended a same target, which significantly complicates computation of marginal target protection. Therefore, our first step is to simplify the pure strategy space to include only those in which all defenders cover different targets. To do so, we create $\mathbf{D}$ *dummy* targets at which rewards and penalties and costs are zero for both defenders and attacker. When the players choose one of these dummy targets, it means they choose to do nothing. As a result, we have $(\mathbf{T} + \mathbf{D})$ targets in total, including these dummy targets. The creation of these dummy targets does not influence the actual outcome of any signaling scheme, but introduce a nice characteristic of the optimal signaling scheme (Lemma 1). This characteristic of at most one defender at each target allows us to provide more efficient algorithms to find an optimal signalling scheme.

**Lemma 1.** *There is an optimal signaling scheme such that for any signal profile $\mathbf{s}$ with a positive probability (i.e., $\omega(\mathbf{s} \mid \lambda) > 0$), then $s(d) \neq s(d')$ for all $d \neq d'$.*

*Proof.* Let's assume in a signaling scheme, there is a signal in which multiple defenders are sent to the same target $t$. We revise this signal by only suggesting the defender $d$ with the lowest cost $C^d(t)$ to $t$ and other defenders are sent to dummy targets instead. First of all, the expected cost will be reduced while the coverage probability at each non-dummy target remains the same. As a result, the principal's objective does not change. Second, the attacker obedience constraints does not change. Third, the LHS of the defender's obedience constraints increases while the RHS is the same. This means no obedience constraint is violated. $\square$

**Step 2:** Working in the dual space. Since LP $(1-4)$ has exponentially many variables, we first reduce it to the following dual linear program $(1-4)$, which turns out to be more tractable to work with:

$$\min \sum_{\lambda} \gamma(\lambda) \text{ s.t.} \tag{5}$$

$$\gamma(\lambda) + \sum_{t'} \left[ U^\lambda(s(-a), t') - U^\lambda(\mathbf{s}) \right] \alpha^\lambda(s(a), t') \tag{6}$$

$$+ q(\lambda) \sum_{d,t'} \left[ U^d(s(-a,-d), t', s(a)) - U^d(\mathbf{s}) \right] \beta^d(s(d), t')$$

$$\geq q(\lambda) U(\mathbf{s}), \forall (\mathbf{s}, \lambda)$$

$$\alpha^\lambda(t, t'), \beta^d(t, t') \geq 0, \forall \lambda, d, t, t'. \tag{7}$$

where each constraint in (6) corresponds to the primal variable $\omega(\mathbf{s} \mid \lambda)$. The dual variables $\alpha^\lambda(t, t')$ correspond to attacker obedience constraints (2). The dual variables $\beta^d(t, t')$ correspond to defender obedience constraints (3). Finally, the variables $\gamma(\lambda)$ corresponds to constraints (4).

Problem (5–7) has an exponential number of constraints. We employ the ellipsoid method [Grötschel et al., 1981] by designing a polynomial-time separation oracle. In this oracle, given a value of $(\alpha^\lambda(t, t'), \beta^d(t, t'), \gamma(\lambda))$, it either establishes that this value is feasible for the problem or, if not, it outputs an hyper-plane separating this value from the feasible region. In the following, we focus on a particular type of oracles: those generating violated constraints. The oracle solves the following optimization problems, each corresponds to a fixed $\lambda$ and $s(a)$ (to be some target $t_0$),

$$\min_{\mathbf{s}:s(a)=t_0} \sum_{t'} \left[ U^\lambda(s(-a), t') - U^\lambda(\mathbf{s}) \right] \alpha^\lambda(t_0, t') \tag{8}$$

$$+ q(\lambda) \sum_{d,t'} \left[ U^d(s(-a,-d), t', t_0) - U^d(\mathbf{s}) \right] \beta^d(s(d), t')$$

$$- q(\lambda) U(\mathbf{s})$$

If the optimal objective of this problem is *strictly* less than $-\gamma(\lambda)$ for any $(\lambda, t_0)$, it means we found a violated constraint corresponding to $(\mathbf{s}^*, \lambda)$ where $\mathbf{s}^*$ is an optimal solution of (8). We iterate over every $(\lambda, t_0)$ to find all violated constraints and add them to the current constraint set.

**Step 3:** Establishing an efficient separation oracle. We now solve (8) for any given $(\lambda, t_0)$. We further divide this problem into two sub-problems; each can be solved via bipartite

matching (which is polynomial-time). More specifically, we divide the signal set $\{\mathbf{s} : s(a) = t_0\}$ into two different subsets, as elaborated in the following.

**Case 1 of Step 3:** Attacked target is not covered. The first subset consists of all signals such that $t_0 \notin s(-a)$, that is, none of the defender is assigned to $t_0$. In this case, the attacker will receive a reward $R^\lambda(t_0)$ for attacking $t_0$ while every defender $d$ receives a penalty $P^d(t_0)$. Thus, each of the following elements in (8) is straightforward to compute:

$$U^\lambda(s(-a), t') - U^\lambda(\mathbf{s}) = \begin{cases} P^\lambda(t') - R^\lambda(t_0) & \text{if } t' \in s(-a) \\ R^\lambda(t') - R^\lambda(t_0) & \text{if } t' \notin s(-a) \end{cases}$$

$$U^d(s(-a,-d), t', t_0) - U^d(\mathbf{s})$$

$$= \begin{cases} C^d(t') - C^d(s(d)) & \text{if } t' \neq t_0 \\ R^d(t_0) + C^d(t_0) - P^d(t_0) - C^d(s(d)) & \text{if } t' = t_0 \end{cases}$$

$$U(\mathbf{s}) = \sum_d P^d(t_0) + C^d(s(d))$$

Given the above computation, we observe that the second and third components (in the second and third lines) of the objective (8), which only depends on the defender utilities, consists of multiple terms — each term depends only on the allocation of each individual defender $(d, s(d))$. On the other hand, the first component (in the first line) of the objective, which depends on the attacker's utility, has terms which depends on targets not in the defender allocation. Therefore, in order to create a corresponding bipartite matching problem, we introduce $|\mathbf{T}|$ new dummy defenders and the following weights between $|\mathbf{T}| + |\mathbf{D}|$ defenders and $|\mathbf{T}| + |\mathbf{D}|$ targets:

$$\eta(d, t) = q(\lambda) \sum_{t' \neq t_0} \left[ C^d(t') - C^d(t) \right] \beta^d(t, t')$$

$$+ q(\lambda) \sum_{t' = t_0} \left[ R^d(t_0) + C^d(t_0) - P^d(t_0) - C^d(t) \right] \beta^d(t, t')$$

$$+ [P^\lambda(t) - R^\lambda(t_0)] \alpha^\lambda(t_0, t) - q(\lambda) C^d(t), \forall t \neq t_0, d \leq |\mathbf{D}|$$

$$\eta(d, t_0) = +\infty, \forall d \leq |\mathbf{D}|$$

$$\eta(d, t) = [R^\lambda(t) - R^\lambda(t_0)] \alpha^\lambda(t_0, t), \forall t, \text{ dummy } d > |\mathbf{D}|$$

Weights associated with these dummy defenders correspond to the terms in (8) which depends on targets not in the actual defender allocation. The weight $\eta(d, t_0) = +\infty$ is to ensure that no actual defender in $\mathbf{D}$ will be assigned to $t_0$.

We now present Lemma 2 (which can be proved via a couple of algebra computation steps), showing that Problem (8) becomes a Minimum Bipartite Matching between $|\mathbf{T}| + |\mathbf{D}|$ defenders and $|\mathbf{T}| + |\mathbf{D}|$ targets.

**Lemma 2.** *The problem (8) can be now reduced to as the following bipartite matching problem using $\eta(d, t)$:*

$$\min_{\mathbf{m}} \sum_d \eta(d, m(d))$$

*after removing the constant term $-q(\lambda) \sum_{d \in \mathbf{D}} P^d(t_0)$ in (8)). Here, $m(d)$ is a target matched to the defender $d$.*

**Case 2 of Step 3:** Attacked target is covered. On other other hand, the second subset consists of all signals such that $t_0$ is assigned to one of the defender. In this case, we further divide this sub-problem into multiple smaller problems, by fixing the defender who covers $t_0$, denoted by $d_0$. Similar to *Sub-problem P1*, we introduce the following weights: $\forall t$

$$\eta(d,t) = q(\lambda) \sum_{t'} [C^d(t') - C^d(t)] \beta^d(t,t')$$
$$+ [P^\lambda(t) - P^\lambda(t_0)] \alpha^\lambda(t_0,t) - q(\lambda) C^d(t), \forall t, \forall d \in \mathbf{D} \setminus \{d_0\}$$
$$\eta(d,t) = [R^\lambda(t) - P^\lambda(t_0)] \alpha^\lambda(t_0,t), \forall t, \text{ dummy } d > |\mathbf{D}|$$

**Lemma 3.** *The problem (8) can be now reduced to as the following bipartite matching problem using $\eta(d,t)$:*

$$\min_{\mathbf{m}} \sum_{d \neq d_0} \eta(d, m(d))$$

*after removing the constant terms $\sum_{t' \neq t_0} [P^{d_0}(t_0) + C^{d_0}(t') - R^{d_0}(t_0) - C^{d_0}(t_0)] \beta^{d_0}(t_0, t') - q(\lambda) \sum_d R^d(t_0)$. In addition, $(d_0, t_0)$ is removed from our matching setting.*

We now have the problem of a Minimum Bipartite Matching between $|\mathbf{T}| + |\mathbf{D}| - 1$ defenders to $|\mathbf{T}| + |\mathbf{D}| - 1$ targets, which can be solved in a polynomial time.

# 4 OPTIMAL EX ANTE PRIVATE SIGNALING

This section relaxes the private signaling requirement and assumes that players make decision on whether to follow signals or not before any signal is sent. Such ex ante private signaling has been studied recently in routing [Castiglioni et al., 2021] and abstract games [Xu, 2020]. However, both works have used the ellipsoid algorithm to compute the optimal scheme. While the ellipsoid algorithm is theoretically efficient, as we will show in our experiments, they are practically quite slow. In our case, we could have also just employed similar technique. However, we take one step further and present a novel idea of using compact representation of the signaling schemes such that the "reduced" signaling space become polynomial size in the number of targets. This important result helps in significantly scaling up the problem computation.

## 4.1 AN EXPONENTIAL-SIZE LP FORMULATION

Overall, the problem of finding an optimal ex ante private signaling scheme can be formulated as the following LP which has an exponential number of variables $\{\omega(\mathbf{s} \mid \lambda)\}$:

$$\max \sum_\lambda q(\lambda) \sum_\mathbf{s} \omega(s \mid \lambda) U(\mathbf{s}) \text{ s.t.} \tag{9}$$

(Attacker obedience) $\forall \lambda, t'$ :

$$\sum_s \omega(\mathbf{s} \mid \lambda) U^\lambda(\mathbf{s}) \geq \sum_\mathbf{s} \omega(\mathbf{s} \mid \lambda) U^\lambda(s(-a), t') \tag{10}$$

(Defender obedience) $\forall d, t'$ :

$$\sum_\lambda q(\lambda) \sum_\mathbf{s} \omega(\mathbf{s} \mid \lambda) U^d(\mathbf{s}) \tag{11}$$
$$\geq \sum_\lambda q(\lambda) \sum_\mathbf{s} \omega(\mathbf{s} \mid \lambda) U^d(s(-d), t', s(a))$$
$$\sum_\mathbf{s} \omega(\mathbf{s} \mid \lambda) = 1, \forall \lambda, \omega(\mathbf{s} \mid \lambda) \geq 0, \forall \mathbf{s}, \lambda \tag{12}$$

Similar to private signaling, we show that the optimal ex-ante signaling scheme can be computed in a polynomial time (Theorem 2) by developing an ellipsoid algorithm.

**Theorem 2.** *The optimal private ex-ante signaling scheme can be computed in a polynomial time.*

## 4.2 COMPACT SIGNALING REPRESENTATION

As we mentioned previously, while the ellipsoid algorithm is theoretically efficient, they run slowly in practice. Therefore, we further show that in this scenario, we can provide a compact representation of signaling schemes such that the signaling space is polynomial in the number of targets. This immediately leads to a polynomial time algorithm for optimal ex ante private signaling by directly solving the polynomial-size linear program. Give any signaling scheme $\omega$, we introduce the new variable $\omega(a \to t, d \to t' \mid \lambda)$ which is the *marginal* probability that the attacker is sent to target $t$ and the defender $d$ is sent to target $t'$, given that the attacker type is $\lambda$. In addition, we introduce $\omega(a \to t \mid \lambda)$ which is the probability the attacker is sent to $t$. Reformulating (9-11) based on these new variables is straightforward. For example, the objective (9) is reformulated as following:

$$\sum_\lambda q(\lambda) \sum_{t,d} \omega(a \to t, d \to t \mid \lambda) R^d(t)$$
$$+ \sum_\lambda q(\lambda) \sum_t \left[ \omega(a \to t \mid \lambda) \right.$$
$$- \sum_d \omega(a \to t, d \to t \mid \lambda)] [\sum_d P^d(t)]$$
$$+ \sum_\lambda q(\lambda) \sum_{t',d} \left[ \sum_t \omega(a \to t, d \to t' \mid \lambda) \right] C^d(t')$$

The crux of this section is the following theorem. It fully characterize the conditions under which the compact representation corresponds to a feasible ex ante signaling scheme.

**Theorem 3.** *The following conditions are necessary and sufficient conditions to generate a feasible ex ante signaling scheme from a compact representation $(\omega(a \to t \mid \lambda), \omega(a \to t, d \to t' \mid \lambda))$:*

$$\sum_t \omega(a \to t \mid \lambda) = 1, \forall \lambda \tag{13}$$
$$\sum_{t'} \omega(a \to t, d \to t' \mid \lambda) = \omega(a \to t \mid \lambda), \forall \lambda, d \tag{14}$$
$$\sum_d \omega(a \to t, d \to t' \mid \lambda) \leq \omega(a \to t \mid \lambda), \forall \lambda, t' \tag{15}$$
$$\omega(a \to t \mid \lambda) \geq 0, \omega(a \to t, d \to t' \mid \lambda) \geq 0, \forall \lambda, t, d, t' \tag{16}$$

*Proof.* It is obvious that these conditions are necessary conditions. Let's consider $\{\omega(a \to t \mid \lambda)\}$ and $\omega(a \to t, d \to t' \mid \lambda)$ satisfying these conditions. We will show that these correspond to a feasible signaling scheme. First, we have:

$$\omega(d \to t' \mid a \to t, \lambda) = \frac{\omega(a \to t, d \to t' \mid \lambda)}{\omega(a \to t \mid \lambda)}$$

which is the probability of assigning defender $d$ to target $t'$ given the attacker is of type $\lambda$ and is assigned to target $t$. By fixing $\lambda$ and $a \to t_0$, we use $\omega(d \to t')$ as an abbreviation of $\omega(d \to t' \mid a \to t_0, \lambda)$ when the context is clear. We will prove that any $\{\omega(d \to t)\}$ satisfying the following conditions correspond to a feasible signaling scheme:

$$\sum\nolimits_t \omega(d \to t) = 1, \forall d$$
$$\sum\nolimits_d \omega(d \to t) \leq 1, \forall t$$

In order to do so, we introduce the following general lemma:

**Lemma 4.** *For any a coverage vector $\{\omega(d,t)\}$ such that:*

$$\sum\nolimits_t \omega(d,t) = r \tag{17}$$
$$\sum\nolimits_d \omega(d,t) \leq r, \tag{18}$$

*given $0 \leq r \leq 1$, there is an assignment of defenders to targets, denoted by $(d_1, t_1), \ldots (d_{|\mathbf{D}|}, t_{|\mathbf{D}|})$, such that:[2]*

- *$\omega(d_i, t_i) > 0$ for all $i \in \{1, \ldots, |\mathbf{D}|\}$*
- *Every maximally-covered target $t$, i.e., $\sum_d \omega(d,t) = r$, is assigned to a defender, that is, $t \in \{t_1, \ldots, t_{|\mathbf{D}|}\}$.*

*Proof.* Let $\mathbf{D}(t) = \{d : \omega(d,t) > 0\}$ the support defender set of target $t$. Similarly, we also denote by $\mathbf{T}(d) = \{t : \omega(d,t) > 0\}$ the support target set of defender $d$. We divide the set of targets into two groups: (i) Group of all maximally-covered targets $\mathbf{T}^{\text{high}} = \{t : \sum_d \omega(d,t) = r\}$; and (ii) Group of other targets $\mathbf{T}^{\text{low}} = \{t : \sum_d \omega(d,t) < r\}$. W.l.o.g, we represent $\mathbf{T}^{\text{high}} = \{t_1, \ldots, t_H\}$ and $\mathbf{T}^{\text{low}} = \{t_{H+1}, \ldots, t_{|\mathbf{T}|+|\mathbf{D}|}\}$ where $\{t_i\}$ is a permutation of targets $\{1, \ldots, |\mathbf{T}| + |\mathbf{D}|\}$.

**Step 1:** Inclusion of high-coverage target group $\mathbf{T}^{\text{high}}$. We first prove that there is a partial allocation from defenders to targets in $\mathbf{T}^{\text{high}}$, denoted by $(d_1, \ldots, d_H)$ such that $d_i \in \mathbf{D}(t)$ for all $t_i \in \mathbf{T}^{\text{high}}$ and they are pair-wise different, i.e., $d_i \neq d_j$ for all $t_i \neq t_j \in \mathbf{T}^{\text{high}}$. We use induction w.r.t $t$.

In the base, $t = 1$, the above statement holds true. Let's assume this statement is true for some $t < |\mathbf{T}^{\text{high}}|$. We will prove that it is also true for $t + 1$. Let's denote by $(d_1, 1), \ldots, (d_t, t)$ the current sequence of defender-to-target assignment. At target $t + 1$, if there is $d \in \mathbf{D}(t+1)$ such that $d \neq d_j$ for all $j \leq t$, then we obtain a new satisfactory partial assignment $\{(d_1, 1), \ldots, (d_t, t), (d, t+1)\}$.

---

[2]Since we have $(\mathbf{T} + \mathbf{D})$ targets in total while there are only $|\mathbf{D}|$ defenders, some targets will not be assigned to any defenders.

Conversely, if $\mathbf{D}(t+1) \subseteq \{d_1, \ldots, d_t\}$, w.l.o.g, we assume $\mathbf{D}(t+1) = \{d_1, d_2, \ldots, d_{t'}\}$ for some $t' \leq t$. We obtain:

**Observation 1.** *There exists a target $t_0 \leq t$ and a defender $d_0 \notin \{d_1, \ldots, d_t\}$ such that $\omega(d_0 \to t_0) > 0$.*

Indeed, if there is no such $(d_0, t_0)$, it means all targets $\{1, \ldots, t+1\}$ can be only assigned to one of the defenders in $\{d_1, \ldots, d_t\}$. As a result, we will have:

$$r \times (t+1) = \sum_{j'=1}^{t+1} \sum_{d \in \mathbf{D}(j')} \omega(d, j') \leq \sum_{j'=1}^{t} \sum_{j'' \in \mathbf{T}(d_{j'})} \omega(d_{j'}, j'')$$
$$= r \times t \text{ (contradiction)} \tag{19}$$

Now, if that target $t_0 \leq t'$, then we obtain a new partial assignment $\{\ldots, (d_0, t_0), \ldots, (d_{t_0}, t+1)\}$ by assigning $d_0$ to target $t_0$ and reallocating $d_{t_0}$ to $t+1$ while keeping other assignments the same. On the other hand, if $t' < t_0 \leq t$, it means $\mathbf{D}(j) \subseteq \{d_1, \ldots, d_t\}$ for all $j \leq t'$. W.l.o.g, let's assume that target $t_0 = t' + 1$. We observe that there must exist a target $t_{00} \in \{1, \ldots, t\} \setminus \{t'+1\}$ and a defender $d_{00} \notin \{d_1, \ldots, d_t\} \setminus \{d_{t'+1}\}$ such that $\omega(d_{00} \to t_{00}) > 0$. Indeed, if there is no such $(d_{00}, t_{00})$, it means all targets $\{1, \ldots, t+1\} \setminus \{t'+1\}$ can be only assigned to one of the defenders in $\{d_1, \ldots, d_t\} \setminus \{d_{t'+1}\}$. As a result, we have:

$$r \times t = \sum_{j'=1, j' \neq t'+1}^{t+1} \sum_{d \in \mathbf{D}_{j'}} \omega(d, j')$$
$$\leq \sum_{j'=1, j' \neq t'+1}^{t} \sum_{j'' \in \mathbf{T}(d_{j'})} \omega(d_{j'}, j'')$$
$$= r \times (t-1) \text{ (contradiction)}$$

Now, if that target $t_{00} \leq t'$ and $d_{00} = d_{t'+1}$, then we can do the swap $(d_{t'+1}, t_{00}), (d_{t_{00}}, t+1), (d_0, t'+1)$ while keeping other assignments the same. If that target $t_{00} \leq t'$ and $d_{00} \neq d_{t'+1}$, then we can do a different swap $(d_{t_{00}}, t+1), (d_{00}, t_{00})$. Finally, if $t_{00} > t'+1$, w.l.o.g, we assume $t_{00} = t' + 2$. We repeated the same above analysis process until at some point, we either already found a feasible assignment or would reach the following situation:

- $\exists d_0 \notin \{d_1, \ldots, d_t\}$ s.t $\omega(d_0 \to t'+1) > 0$
- $\exists d_{00} \notin \{d_1, \ldots, d_t\} \setminus \{d_{t'+1}\}$ s.t. $\omega(d_{00} \to t'+2) > 0$
- $\ldots$
- $\exists d_{\text{final}} \notin \{d_1, \ldots, d_{t'}\}$ and $\exists t_{\text{final}} \in \{1, \ldots, t'\}$ such that $\omega(d_{\text{final}} \to t_{\text{final}}) > 0$ where final $= [0]^{t-t'}$.

In this situation, we first swap $(d_{t_{\text{final}}}, t+1), (d_{\text{final}}, t_{\text{final}})$. There are two cases. If $d_{\text{final}} \notin \{d_1, \ldots, d_t\}$, then we found a solution. If $d_{\text{final}}$ is equal to some $d_{t'+j}$ for some $j \leq t - t'$, we then reassign $(d_{[0]^{t'+j}}, t'+j)$. At this step, there are two cases again. That is either $d_{[0]^{t'+j}} \notin \{d_1, \ldots, d_t\}$ or $d_{[0]^{t'+j}}$

is one of $\{d_{t'+1}, \ldots, d_{t'+j-1}\}$. The former case means we found a solution while the latter case indicates we have to do the reassignment again for a target in $\{t'+1, \ldots, t'+j-1\}$. Observe that, every time we have to do a reassignment, the index of the target for the reassignment is decreased. In the end, it will reach target $t'+1$ for which we can reassign $d_0 \notin \{d_1, \ldots, d_t\}$ and obtain a feasible solution.

**Step 2:** Extension to include target group $\mathbf{T}^{\text{low}}$. We are going to prove that there is an assignment from defenders $\mathbf{D}$ to $|\mathbf{D}|$ targets, including all targets in $\mathbf{T}^{\text{high}}$. We apply induction with respect to the defender $d$. Note that we cannot apply induction with respect to the targets $t$ since we include target group $\mathbf{T}^{low}$ in this analysis and as a result, the equality on the LHS of (19) no longer holds.

In the base, we start with the feasible assignment of the group $\mathbf{T}^{\text{high}}$. Then at each induction step, we perform a defender-target swapping process which is similar to the case of high-coverage target group $\mathbf{T}^{\text{high}}$. The tricky part is that for any swapping, we do not get rid of any targets that have been assigned so far (besides changing the defender assigned to them). It means that in the final assignment of the induction process, denoted by $(1, t_1), \ldots, (|\mathbf{D}|, t_{|\mathbf{D}|})$, all targets in $\mathbf{T}^{\text{high}}$ are still included. $\square$

Based on the result of Lemma 4, we allocate the following non-zero probability to the assignment with $r = 1$:

$$p = \min\{\min_d\{\omega(d, t_d)\}, r - \max_{t \notin \{t_1, \ldots, t_{\mathbf{D}}\}} \sum_d \omega(d, t)\}$$

Given this assignment, we update $w(d, t_d) = w(d, t_d) - p$ for all $d$. The resulting coverage vector $\{\omega(d, t)\}$ still satisfies the conditions (17–18) with the remaining $r = r - p < 1$. We keep doing this probability allocation until we obtain a feasible signaling scheme (aka, $r$ reaches 0). $\square$

## 5 EXPERIMENTS

In our experiments, we aim at evaluating both the solution quality and runtime performance of our algorithms in various game settings. All the LPs in our algorithms are solved with the CPLEX solver (version 20.1). We run our algorithms on a machine with an Intel i7-8550U CPU and 15.5GB memory. The rewards and penalties of players are generated uniformly at random between [0, 20] and [-20, 0], respectively. All data points are averaged over 40 random games and the error bars represents the standard error.

We compare our private and ex ante signaling schemes with: (i) a *baseline* method in which each defender optimizes his utilities separately by solving a Bayesian Stackelberg equilibrium between that defender and attacker, without considering strategies of other defenders; and (ii) the Nash Stackelberg equilibrium (NSE) among the defenders. We use the method provided in [Gan et al., 2018] to approximate

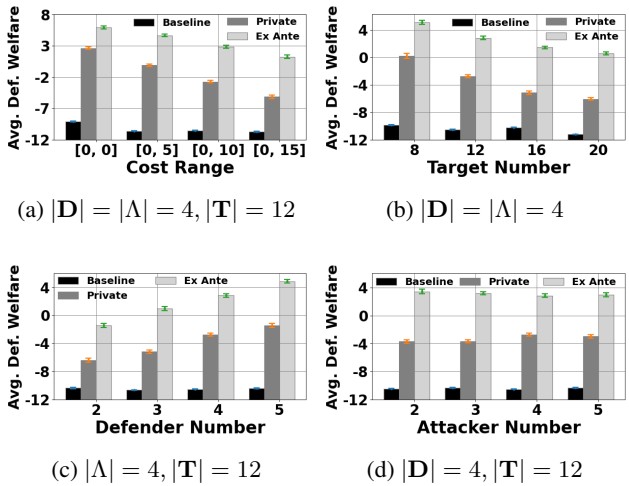

Figure 1: Average Defender Social Welfare. The defenders' cost range is fixed to $[0, 10]$ in sub-figures (b), (c), (d).

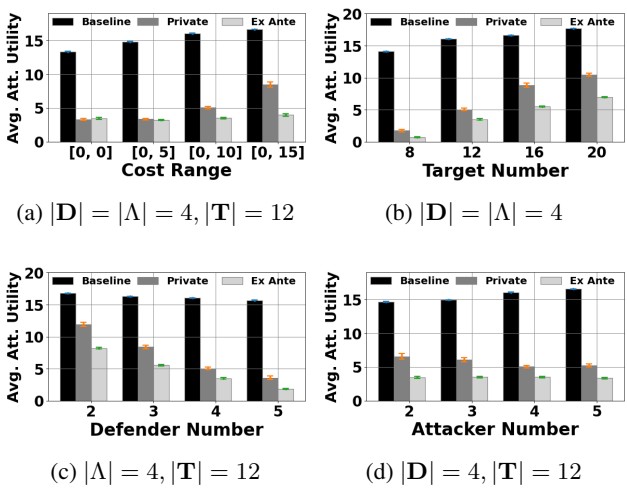

Figure 2: Average Attacker Utility. The defenders' cost range is fixed to $[0, 10]$ in sub-figures (b), (c), (d).

an NSE. We evaluate our signaling schemes in two scenarios corresponding to two different objectives of the principal: (i) maximizing the social welfare of the defenders (Figures 1–3); and (ii) maximizing her own defense utility (i.e., the principal is one of the self-interested defenders) (Figure 4). Next, we highlight our important results.

In Figures 1 and 2, the x-axis is either the defender's cost range (the defense cost of each defender is randomly generated within this range), or the number of targets, or the number of defenders, or the number of attacker types. The y-axis is either the defender social welfare (Figure 1) or the average utility of the attacker (Figure 1). Note that in these figures, we do not consider the Nash Stackelberg equilibrium (NSE) among the defenders. This is because the method provided in [Gan et al., 2018] to approximate an NSE is only applicable for the no-patrolling-cost setting. Figure 1 shows that

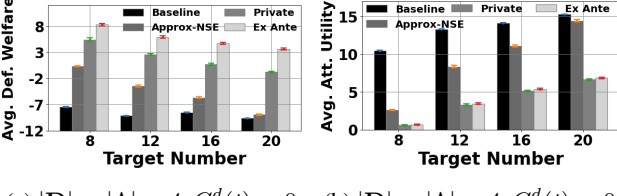

(a) $|\mathbf{D}| = |\Lambda| = 4, C^d(t) = 0$  (b) $|\mathbf{D}| = |\Lambda| = 4, C^d(t) = 0$

Figure 3: All evaluated algorithms, no patrolling costs.

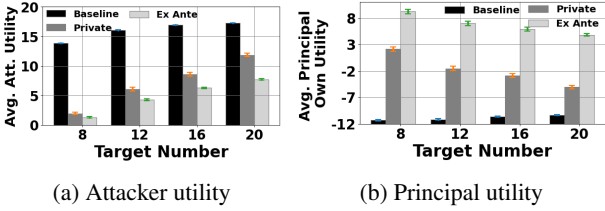

(a) Attacker utility  (b) Principal utility

Figure 4: The principal optimizes her own utility when $|\mathbf{D}| = |\Lambda| = 4$ and the defenders' cost range $C^d(t) \in [0, 10]$.

signaling schemes (`Private` and `ExAnte`) helps in significantly increasing the defender social welfare compared to the `Baseline` case. In addition, the defender social welfare in `ExAnte` is substantially higher than the `Private` case. This result makes sense since the persuasion constraints in `ExAnte` are less restricted. In addition, the social welfare is roughly a decreasing linear function of the cost range and the number of targets while it increases linearly in the number of defenders. This is because the social welfare is a decreasing function of the defenders' coverage probability at each target and the higher the number of defenders is, the more coverage at each target is. Conversely, we see an opposite trend in the attacker graphs (Figure 2).

Furthermore, we include the NSE in our experiments with no defense cost. Figure 3 shows that despite `NashStackelberg` results in a higher social welfare for the defenders compared to `Baseline` in which each defender ignores the presence of other defenders, the social welfare in `NashStackelberg` is still significantly lower than `Private` and `ExAnte`. The results in Figures 1, 2 and 3 clearly show that coordinating the defenders through the principal's signaling schemes helps in significantly enhancing the protection effectiveness on the targets.

In Figure 4, we examine the situation in which the principal attempts to maximize her own utility (given she is one of the self-interested defenders). We again observe that the attacker suffers a significant loss in its utility compared to `Baseline` (Figure 4(a), `Private` and `ExAnte` versus `Baseline`). Conversely, the principal can get a significant benefit for strategically revealing her private information through the signaling mechanisms (Figure 4(b)).

Figure 5 shows the logarithm runtime of our algorithms compared to `Baseline` and NSE. We observe that our algorithms

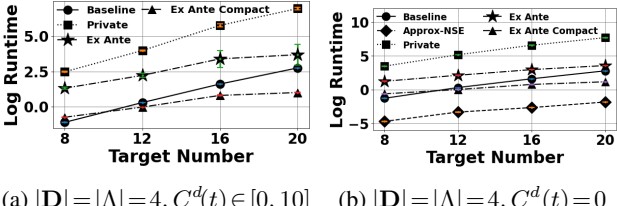

(a) $|\mathbf{D}| = |\Lambda| = 4, C^d(t) \in [0, 10]$  (b) $|\mathbf{D}| = |\Lambda| = 4, C^d(t) = 0$

Figure 5: Log run time in seconds.

(`Private` and `ExAnte`) are suitable for medium games. In Figure 5(a), it takes `Private` and `ExAnte` approximately 23 minutes and 40 seconds respectively to solve 20-target games. Furthermore, our compact representation method (`ExAnteCompact`) helps in solving the signaling scheme significantly faster. It only takes `ExAnteCompact` approximately 2.7 seconds to solve 20-target games.

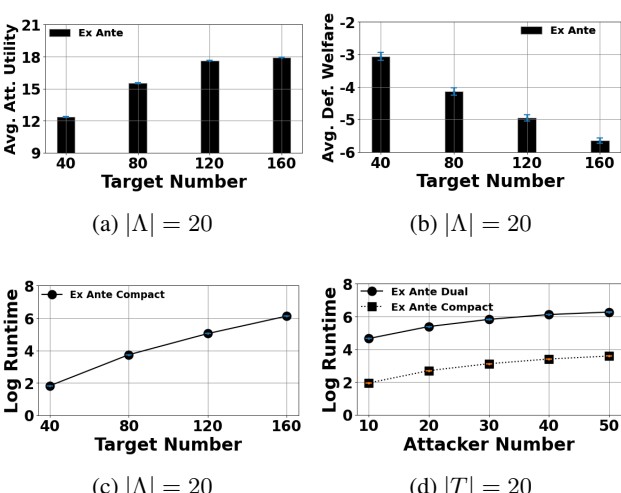

(a) $|\Lambda| = 20$  (b) $|\Lambda| = 20$

(c) $|\Lambda| = 20$  (d) $|T| = 20$

Figure 6: Scalability of target number or attacker types in ex ante setting when $|D| = 4$ and the defenders' cost range $C^d(t) \in [0, 10]$.

Finally, we examine the performance in the ex ante case with large number of targets or attacker types in Figure 6. Our algorithms can easily scale to about 160 targets, which makes large improvement compared to previous works [Yin and Tambe, 2012, Nguyen et al., 2014]. We remark that it is typically impossible to test running time for such complicated security games for more than 200 targets on a single machine (most real world applications such as conservation area protection or border protection have less than 200 targets as well). For large number of attacker types, our experiments show that the running time dependence of our algorithm on the number of attacker types is linear, which is extremely efficiency.

# 6 SUMMARY

In this paper, we study information design in a Bayesian security game setting with multiple independent and self-interested defenders. Our results (both theoretically and empirically) show that information design not only significantly improves protection effectiveness but also leads to efficient computation. In particular, in computing an optimal private signaling scheme, we develop an ellipsoid-based algorithm in which the separation oracle component can be decomposed into a polynomial number of sub-problems, and each sub-problem reduces to a bipartite matching problem. This is a non-trivial task, since the outcomes of private signaling form the set of Bayes correlated equilibria and computing an optimal correlated equilibrium is a fundamental and well-known intractable problem. Our proof is technical and crucially explores the special structure of security games. Furthermore, we investigate the *ex-ante* private signaling scheme. In this scenario, we develop a novel compact representation for the signaling schemes by compactly characterizing jointly feasible marginals. This finding enables us to significantly reduce the signaling scheme computation compared to the ellipsoid approach (which is efficient in theory but slow in practice).

**Acknowledgements**

Haifeng Xu is supported by an NSF grant CCF-2132506; this work is done while Xu is at the University of Virginia. Thanh H. Nguyen is supported by ARO grant W911NF-20-1-0344 from the US Army Research Office.

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
