# OpenReview forum: "Information Design for Multiple Independent and Self-Interested Defenders: Work Less, Pay Off More"
_auai.org/UAI/2022/Conference — UAI 2022 Poster_

### Official Review · Reviewer_FJkv · 2022-04-10

**Q2(1) Originality/Novelty:** 3
**Q2(2) Significance/Impact:** 3
**Q2(3) Correctness/Technical Quality:** 3
**Q2(6) Clarity Of Writing:** 3
**Q6 Overall Score:** 7
**Q8 Confidence In Your Score:** 4

**Q1 Summary And Contributions:**

The paper deals with the problem of information design for private signalling in a setting of multi-defender security games. A polynomial-time algorithm is presented to compute an optimal private signalling scheme and a compact representation of ex-ante persuasive signalling scheme is proposed that enable an efficient computation of an optimal scheme. Moreover, extensive results are presented to validate the theoretical results.

**Q2 Assessment Of The Paper:**

More detailed information regarding each of these aspects is given below:

**Q2(4) Quality Of Experiments (Optional):**

3: Good: The experimental evaluation is adequate, and the results convincingly support the main claims.

**Q2(5) Reproducibility:**

3: Good: Key resources (e.g., proofs, code, data) are available and key details (e.g., proofs, experimental setup) are sufficiently well-described for competent researchers to confidently reproduce the main results.

**Q3 Main Strengths:**

Security games are a very-well studied class of games that have a lot of real-world applications and a high impact for the AI community.
The paper applies Information Design techniques to multi-defender security games. The main result of the paper is a polynomial-time ellipsoid algorithm to compute an optimal private signalling scheme. This is obtained by reducing the separation oracle of the ellipsoid algorithm to a polynomial number of bipartite matching instances. Moreover, for the restricted case of ex-ante persuasive signalling schemes, where players decide in advance whether to follow the indications of the signalling scheme, a compact representation of the signalling scheme is designed that allows to significantly reduce the time needed to compute the optimal scheme.

Technical results are very interesting and not trivial. Experimental data are discussed to evaluate both the quality of the solution provided by the algorithm and its runtime performances.


**Q4 Main Weakness:**

The description of the results is very technical and hard to follow to readers not confident with the field. Maybe, some intuitions could help in increasing the readability.

**Q5 Detailed Comments To The Authors:**

In the Information Design framework the principal wants to minimize the information revealed to other players. Even if it is easy to imagine why the revelation has to bee maintained low it could be helpful to argument why this happens in the setting of your running example of wildlife conservation.

Authors say that their proof crucially explores the special structure of security games. Which are the specific characteristics that you exploited and is it possible to characterize a class of games where your algorithm can be applied.

Is it realistic to assume that when an attacker is captured all the defenders receive the same reward? Maybe, there are two kinds of rewards: a general one, that is common to all the defenders, that has this attacker eliminated; a specific one, that could be a monetary reward obtained by the team that makes the  capture. Similarly, since defenders have different targets, the penalty suffered by defenders when an attacker hits a target could be different. This scenario could give rise to a "combinatorial" version of multi-defender security games. Could your Information Design approach effective also in this case?

**Q7 Justification For Your Score:**

Multi-defender security games are an important class of games with a lot of real-world applications. Information Design applied in this setting could pave the way to several applications.

Results presented in the paper are interesting and not trivial.  Theoretical results are supported by extensive experimental data related to both the quality of the solution computed by the algorithm and the runtime performance.

My only issue is about the presentation that is too technical.

**Q9 Complying With Reviewing Instructions:**

1: Yes.

---

### Official Review · Reviewer_NMnb · 2022-04-12

**Q2(1) Originality/Novelty:** 3
**Q2(2) Significance/Impact:** 2
**Q2(3) Correctness/Technical Quality:** 3
**Q2(6) Clarity Of Writing:** 4
**Q6 Overall Score:** 6
**Q8 Confidence In Your Score:** 2

**Q1 Summary And Contributions:**

This work solves the multi-defender security games using information design. They build a signaling scheme for the principle (a privileged defender with private information) to influence the decision of other defenders and the attacker by strategically revealing their private information. The experiments show that this signaling approach increases the protection while improving the computational efficacy. They tested against individual strategies for each defender and against NSE strategies.

**Q2 Assessment Of The Paper:**

More detailed information regarding each of these aspects is given below:

**Q2(4) Quality Of Experiments (Optional):**

2: Fair: The experimental evaluation is weak: important baselines are missing, or the results do not adequately support the main claims.

**Q2(5) Reproducibility:**

3: Good: Key resources (e.g., proofs, code, data) are available and key details (e.g., proofs, experimental setup) are sufficiently well-described for competent researchers to confidently reproduce the main results.

**Q3 Main Strengths:**

The paper tackles the problem of having independent, not explicitly coordinating defenders protecting a set of targets in the presence of adversaries in the system. This problem is called a multi-defender security game and it has been investigated in literature mainly using pure individual strategies for the defenders without a coordination protocol which hurts the protection objective. On the other hand, the solutions that tried to converge into a coordination strategy using the Nash-Stackelberg equilibrium have been proved to be NP-hard when considering the adversary's response plans. This paper takes a different approach to use information design to build an implicit signaling scheme which has been proven in the experiments to effectively achieve a higher degree of protection.

**Q4 Main Weakness:**

 The paper didn't consider the learning-based solutions for this problem in the related work or the experiment section. This problem has been solved using decentralized-training and decentralized-execution MARL algorithm and it would be good to compare the proposed approach with MARL approach in terms of scalability, applicability, efficiency, and generalizability.

**Q5 Detailed Comments To The Authors:**

1) Can explain the generalizability of your approach in terms of the problem domain, number of defenders and attackers?
2) How does the proposed approach scale in terms of the number of attackers?
3) How would differentiate your approach from those developed based on MARL as [2020-Learning to Communicate Implicitly by Actions] and [2019-Social Influence as Intrinsic Motivation for Multi-Agent Deep Reinforcement Learning]? Those approaches deal with similar but broader problems using an implicit signaling scheme through the agents' actions. I understand you are having a different approach from learning but it would be valuable to the reader if you can discuss the pros and cons of both directions.
4) Please consider reducing the number of sentences in bold font.

**Q7 Justification For Your Score:**

The paper takes a new direction in solving the multi-defender security games by adopting the information design theory. It also gives very detailed proofs to their idea and assumptions.

**Q9 Complying With Reviewing Instructions:**

1: Yes.

---

### Official Review · Reviewer_HhbG · 2022-04-14

**Q2(1) Originality/Novelty:** 2
**Q2(2) Significance/Impact:** 2
**Q2(3) Correctness/Technical Quality:** 2
**Q2(6) Clarity Of Writing:** 2
**Q6 Overall Score:** 5
**Q8 Confidence In Your Score:** 2

**Q1 Summary And Contributions:**

This paper studies the information design in a Bayesian security game setting with multiple independent and self-interested defenders. It  theoretically and empirically shows that information design not only significantly improves protection effectiveness but also leads to efficient computation. The experiment results validated the proposed approach.

**Q2 Assessment Of The Paper:**

More detailed information regarding each of these aspects is given below:

**Q2(4) Quality Of Experiments (Optional):**

2: Fair: The experimental evaluation is weak: important baselines are missing, or the results do not adequately support the main claims.

**Q2(5) Reproducibility:**

2: Fair: Key resources (e.g., proofs, code, data) are unavailable but key details (e.g., proof sketches, experimental setup) are sufficiently well-described for an expert to confidently reproduce the main results.

**Q3 Main Strengths:**

1. This paper develop an ellipsoid-based algorithm in which the separation oracle component can be decomposed into a polynomial number of sub-problems, and each sub-problem reduces to a bipartite matching problem. This improves the practical solving speed of this problem.

2. The experiment results validate the proposed methods.

**Q4 Main Weakness:**

1. The paper claims this problem can be solved in polynomial time. However, optimal correlated equilibrium is proved to be NP-hard in many succinct games. It is important to list the assumptions made in this paper.

2.The experiment results is limited to small number of targets, K=20, it will be interesting to see results when K is larger than 1000.

**Q5 Detailed Comments To The Authors:**

1. List assumptions of this paper which leads to polynomial solvable.

2. Add more experiments when number of targets K is large enough.

**Q7 Justification For Your Score:**

This paper develop an ellipsoid-based algorithm in which the separation oracle component can be decomposed into a polynomial number of sub-problems, and experiment results validates the effectiveness of proposed method. The conclusion looks like interesting to this area.

**Q9 Complying With Reviewing Instructions:**

1: Yes.

---

### Decision · Program_Chairs · 2022-05-15

**Decision:**

Accept (Poster)

**Comment:**

Meta Review: The reviewers agree that the authors study an interesting problem and provide and technically strong and novel solution.